# Type I interferons and MAVS signaling are necessary for tissue resident memory CD8+ T cell responses to RSV infection

Augusto Varese [ID][¤], Joy Nakawesi, Ana Farias, Freja C. M. Kirsebom [ID], Michelle Paulsen [ID], Rinat Nuriev, Cecilia Johansson [ID]*

Respiratory Infections Section, St Mary's campus, National Heart and Lung Institute, Imperial College London, London, United Kingdom

¤ Current address: Instituto de Investigaciones Biomédicas en Retrovirus y SIDA (INBIRS), Facultad de Medicina Universidad de Buenos Aires, Buenos Aires, Argentina
* c.johansson@imperial.ac.uk

**Data Availability Statement:** All relevant data are within the manuscript and its Supporting Information files.

## Abstract

Respiratory syncytial virus (RSV) can cause bronchiolitis and viral pneumonia in young children and the elderly. Lack of vaccines and recurrence of RSV infection indicate the difficulty in eliciting protective memory immune responses. Tissue resident memory T cells (T$_{RM}$) can confer protection from pathogen re-infection and, in human experimental RSV infection, the presence of lung CD8+ T$_{RM}$ cells correlates with a better outcome. However, the requirements for generating and maintaining lung T$_{RM}$ cells during RSV infection are not fully understood. Here, we use mouse models to assess the impact of innate immune response determinants in the generation and subsequent expansion of the T$_{RM}$ cell pool during RSV infection. We show that CD8+ T$_{RM}$ cells expand independently from systemic CD8+ T cells after RSV re-infection. Re-infected MAVS and MyD88/TRIF deficient mice, lacking key components involved in innate immune recognition of RSV and induction of type I interferons (IFN-α/β), display impaired expansion of CD8+ T$_{RM}$ cells and reduction in antigen specific production of granzyme B and IFN-γ. IFN-α treatment of MAVS deficient mice during primary RSV infection restored T$_{RM}$ cell expansion upon re-challenge but failed to recover T$_{RM}$ cell functionality. Our data reveal how innate immunity, including the axis controlling type I IFN induction, instructs and regulates CD8+ T$_{RM}$ cell responses to RSV infection, suggesting possible mechanisms for therapeutic intervention.

## Author summary

Lung resident memory CD8+ T cells are an important component of the memory immune response to respiratory pathogens. They are located within the lungs and provide frontline defense against re-infection by rapidly responding, proliferating and eliminating infected cells to limit viral spread. Here, we show that the innate immune response during Respiratory Syncytial Virus (RSV) infection determines how lung resident CD8+ memory T cells behave during a re-encounter. We identify a host component of the early response to the

**Funding:** A.V. was supported by a European Respiratory Society and the Asociación Latinoamericana de Tórax joint long-term Research Fellowship 2019 (Number: LTRF 201901-00546; https://www.ersnet.org), F.C.M.K. was supported by a PhD Fellowship from the Wellcome Trust (109058/Z/15/Z; https://wellcome.org) and M.P. was supported by a PhD Fellowship from the National Heart and Lung Institute Foundation (registered charity number 1048073; https://www.nhlf.info). R.N. was supported by a fellowship from the UK-Russia Young Medics Association supported by Sechenov First Moscow State Medical University and the British Embassy Moscow. C.J. was supported by a grant from the Medical Research Council (Grant G0800311 and MR/V000659/1; https://mrc.ukri.org), Rosetrees Trust and Stoneygate Trust (M370 and M370-F1; https://rosetreestrust.co.uk) and CRUK (A27217; https://www.cancerresearchuk.org). This research was funded in whole, or in part, by the Wellcome Trust [109058/Z/15/Z]. For the purpose of open access, the author has applied a CC BY public copyright licence to any Author Accepted Manuscript version arising from this submission. The sponsors or funders did not play any role in the study design, data collection and analysis, decision to publish, or preparation of the manuscript.

**Competing interests:** The authors have declared that no competing interests exist.

virus–type I interferons–that is required for adequate expansion and function of lung resident CD8+ memory T cells. These studies uncover important innate immune regulatory mechanisms of the memory T cell response to viral infection and may have application in vaccination and therapy.

## Introduction

The lungs are a major gateway for highly contagious pathogens that constantly threaten human health. As respiratory viral infections are usually confined to the lung and only spread systemically in some of the most severe cases, control of infection mostly relies on lung resident immune mechanisms [1]. However, many viral pathogens successfully evade the induction of durable and effective memory responses in the lung after natural infection or vaccination and are therefore able to cause multiple infections throughout a person's lifetime [2–4]. Complementary to antibodies, lung tissue resident memory T cells ($T_{RM}$) are increasingly appreciated as a key component of protective responses in the lung [5]. $T_{RM}$ cells are subsets of CD8+ and CD4+ T cells that reside in peripheral tissues. Compared to their circulatory counterparts, $T_{RM}$ cells are in place and poised to rapidly respond to antigen stimulation and/ or inflammatory mediators released during a second encounter with the pathogen [6]. CD8+ $T_{RM}$ cells are defined by co-expression of CD69 and integrin $\alpha_E$ sub-unit (CD103) [7]. In addition, in the lungs of humans and mice, $T_{RM}$ cells can also express integrin $\alpha_1$ sub-unit (CD49a) and CCL16 chemokine receptor CXCR6[8–10] but are heterogeneous and exhibit different expression levels of other surface markers [11,12]. In the lungs, $T_{RM}$ cells are important for the control of viral respiratory infections, as well as being the main drivers of heterosubtypic immunity to influenza A (IAV) infection [13–19].

Human respiratory syncytial virus (henceforth RSV) is a major cause of lower respiratory tract infections, accounting for 33.1 million new cases each year in children younger than five years and resulting in 3.2 million hospitalizations and as many as 118.200 deaths globally [4]. In addition, several reports show associations between pediatric RSV infection and recurrent wheeze in school years, as well as possible links to the development of asthma in adulthood [20,21]. Furthermore, RSV is also causing severe disease in the elderly and mortality caused by RSV infection in over 65-year-old adults is estimated to be 7.2 of 100.000 persons/year in the USA [22].

A study in which healthy adult volunteers were experimentally challenged with RSV showed that the abundance of pre-existing RSV-specific respiratory CD8+ $T_{RM}$ cells prior to infection strongly correlated with reduced symptoms and decreased viral load [17]. Of note, it has been proposed that reduced levels of $T_{RM}$ cells formed in the lung of neonatal mice correlate with increased susceptibility to IAV re-infection [23]. Despite their relevance, CD8+ $T_{RM}$ cells in the lung (in contrast to other mucosal sites) are relatively short-lived and it is unclear what combination of factors is necessary to induce and maintain their numbers and activity [24,25].

Innate immunity is important to control virus infections and for activation of the adaptive immune response. The innate immune response is initiated when the pathogens are detected by pattern recognition receptors (PRRs). RNA viruses such as RSV, IAV and coronaviruses are sensed via endosomal PRRs of the Toll-like receptor family (TLR-3, 7 and 8) [26]. Engagement of these receptors leads to a signaling cascade mediated by myeloid differentiation-primary response 88 (MyD88) and TIR-domain-containing adapter-inducing interferon-β (TRIF) adaptor proteins [27]. In parallel, RNA viruses are also sensed via cytosolic PRRs, mainly

retinoic acid-inducible gene I (RIG-I) and melanoma differentiation-associated protein 5 (MDA-5) [28–30]. Signaling from both these receptors converges on the protein mitochondrial antiviral signaling (MAVS) to induce expression of type I IFNs and other pro-inflammatory cytokines [30,31]. We have previously reported that after RSV recognition, type I IFNs produced by alveolar macrophages (AMs) ignite the early innate response in a MAVS-dependent fashion [32]. As a result, MAVS deficient (*Mavs*$^{-/-}$) mice fail to control early RSV replication and show heightened weight loss before recovery [32]. Virus-derived RNA can also be sensed by endosomal TLR3, TLR7 and TLR8 that signal via MyD88 and/or TRIF adaptor proteins [27]. In the RSV murine infection model, genetic deletion of MyD88 and TRIF (*Myd88/Trif*$^{-/-}$) does not affect viral control and disease progression, in contrast to what happens in *Mavs*$^{-/-}$ mice [33,34]. However, *Myd88/Trif*$^{-/-}$ mice fail to mimic the full extent of wild type (wt) immune response to RSV and show impaired neutrophil recruitment to the lungs [33].

Type I IFNs are the main drivers of early immune responses against viral infections in the lung [35,36]. However, type I IFNs also play a role in T cell differentiation and memory program acquisition [37] and T cells cultured *ex vivo* in the presence of type I IFNs upregulate residency-associated markers [38]. We and others have reported that during the adaptive phase of the immune response against primary RSV infection, *Mavs*$^{-/-}$ mice are able to elicit RSV-specific CD4+ and CD8+ T cell responses comparable to those in wt mice [32,34,39]. However, *Mavs*$^{-/-}$ mice exhibit an impaired memory response to RSV, displaying reduced numbers of RSV-specific CD8+ T cells [39]. Whether type I IFNs, MAVS and MyD88/TRIF signaling also impact $T_{RM}$ cell responses in the lung has not been previously addressed. In this report, we show that RSV re-infection leads to a drastic expansion of lung resident $T_{RM}$ cells displaying effector function that is markedly impaired in MAVS (*Mavs*$^{-/-}$) and MyD88/TRIF (*Myd88/Trif*$^{-/-}$) deficient mice. Notably, administration of IFN-α to *Mavs*$^{-/-}$ mice during the early phase of primary infection restored $T_{RM}$ cell expansion but not functionality during secondary infection. These results establish a link between innate immunity, particularly type I IFN responses, and CD8+ lung $T_{RM}$ cells after RSV re-infection, which may impact development of better RSV vaccines and therapies.

## Results

### Lung $T_{RM}$ cells expand post RSV re-infection

Tissue resident T cells seed the lung after primary infection and are maintained locally at low numbers to rapidly react to a re-infection. To assess this in mice, C57BL/6 mice were intranasally (i.n.) infected with RSV. After 25 days, CD8+ effector T cells, in particular ones expressing CD69 and CD103 (corresponding to $T_{RM}$ cells), were detected in the lung (Fig 1A–1C). To study the dynamics of the response, we looked at earlier times after re-challenge. Mice at day 21 post primary infection were re-challenged i.n. with the same virus. Lung tissue and bronchoalveolar fluid (BAL; containing cells from the airways) were obtained at days 1, 2, 3 and 4 post re-infection, and key immune cell populations were identified by flow cytometry (for gating strategy see S1A and S1B Fig). The number of total leukocytes (CD45+) cells increased at days 2–4. One day after re-infection, neutrophil numbers peaked in the airways and lung tissue while alveolar macrophage (AM) numbers remained overall unchanged during the re-infection period (S2A–S2C Fig). These responses are very similar to what is observed during primary infection [32,33] but to a much lower magnitude. Naïve CD4+ and CD8+ T cell numbers in the lung remained constant, yet as expected, total CD8+ and CD4+ T cell numbers increased during re-infection (S2D–S2G Fig). The numbers of total CD4+ and CD8+ T cells at day 4 post re-infection were comparable to numbers recorded at the peak of the primary infection (day 7–8) in this model [32,39,40]. The increase was driven mainly by an expansion in the number

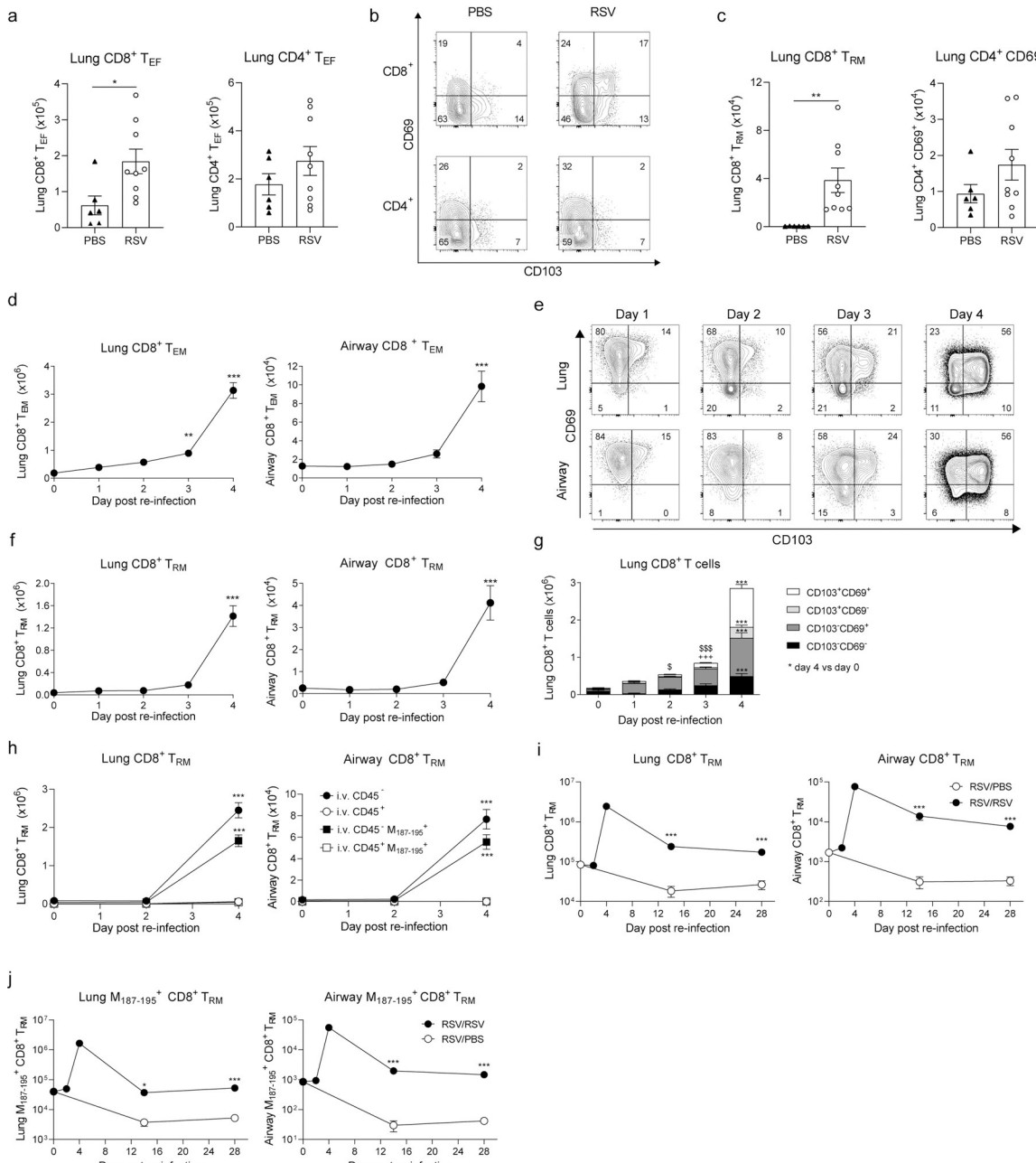

**Fig 1. Tissue resident memory T cells ($T_{RM}$) in the lungs following RSV secondary challenge.** Mice were mock (PBS) or infected with RSV i.n and 25 days later lung cells were recovered to assess CD8+ and CD4+ T cell populations. (**a**) Total number of CD8+ and CD4+ effector T cells ($T_{EF}$; CD62L- CD44+). (**b**) Representative flow cytometry plots showing CD103 and CD69 expression in CD8+ and CD4+ $T_{EF}$ cells. (**c**) Total number of CD8+ $T_{RM}$ cells (CD103+ CD69+) and CD69+ CD4+ T cells. Data are presented as mean±SEM of 6 PBS and 9 RSV mice pooled from two independent experiments. Mice were infected with RSV i.n. and at 3 weeks p.i. mice were mock or re-challenged with RSV and lung and airway cells were analyzed at the indicated time points by flow cytometry (day 0 = mock re-infection). (**d**) Total number of lung and airway CD8+ $T_{EF}$ cells (CD62L- CD44+) cells. (**e**) Representative flow cytometry plots for CD69 and CD103 expression on lung and airway CD8+ $T_{EF}$ cells. (**f**) Total number of CD69+ CD103+ $T_{RM}$ cells were quantified in lung and airway. (**g**) Stacked analysis of the subpopulations of CD8+ $T_{EF}$ cells bases on expression of CD103 and/or CD69. RSV re-infected mice were intravenously (iv.) injected with α-CD45-BUV395 10 min before euthanasia and lung and airway (**h**) CD8+ $T_{RM}$ cells and $M_{187-195}$ specific $T_{RM}$ cells were quantified in iv. stained (i.v. CD45+) and unstained (i.v. CD45-) cells. RSV re-infected mice were culled 14 and 28 days post re-infection and i.v.- $T_{RM}$ cells and $M_{187-195}$ specific $T_{RM}$ cells were quantified in (**i**) lung and (**j**) airways. Data are presented as the mean±SEM of 9–11 (RSV/RSV) individual mice per time point, pooled from two independent experiments. Statistical significance of differences between day 0 (mock re-infection) and other time points was determined by one-way ANOVA with Tukey's post hoc test and indicated as *. In panel g $ represents differences within CD69+ CD103- population and + represent differences within CD69- CD103-. * P ≤ 0.05, ** P ≤ 0.01, *** P ≤ 0.001.

of effector CD8+ and CD4+ T cells (CD62L- CD44+) in both the lung and in BAL (Figs 1D and S2F). Classically defined CD8+ CD103+ CD69+ $T_{RM}$ cells in the lung and BAL were analyzed (Fig 1E) and found to be a major component of lung and BAL at 4 days after re-infection (Fig 1F). A more detailed analysis of CD69 and CD103 expression revealed that at day 1 post re-infection there was an increase in CD69+ CD103- CD8+ T cells and between days 3–4 both CD69+ CD103- and CD69+ CD103+ CD8+ T cells expanded in the lung (Fig 1G). Of note, the same dynamic was observed in CD69 expressing CD4+ T cells, with a remarkable increase by day 4 post re-infection both in lung and airways (S2H Fig). To confirm that CD69+CD103+ T cells that expand after RSV re-infection are localized within the tissue and not the vasculature, CD45+ cells in circulation were intravenously labeled 10 minutes before euthanasia with BUV395 conjugated anti-CD45 antibody (CD45 i.v.) and $T_{RM}$ cells were analyzed at days 0, 2 and 4 post re-infection (gating strategy S1C Fig). Lung and airway CD69+CD103+ CD8+ T cells were i.v.-, i.e., not binding anti-CD45 in the circulation, and peaked at day 4 post re-infection (Fig 1H). To analyze RSV-specific CD8+ T cells, they were labeled using $M_{187-195}$ peptide loaded tetramers (gating strategy S1D Fig). Both lung and airway $M_{187-195}$ specific $T_{RM}$ cells were found in the i.v.CD45- fraction and followed the same expansion kinetics as total $T_{RM}$ cells, peaking at 4 days post-re-challenge (Fig 1H) but remaining detectable at least up to 28 days post RSV re-infection (Fig 1I and 1J). To confirm that expansion of $T_{RM}$ cells during re-infection is driven by specific RSV recognition, RSV-infected mice were infected 3 weeks later with an unrelated viral pathogen, influenza A virus (X31 strain). At day 4 post infection, lung and airway $T_{RM}$ cells (total or RSV-specific) did not expand in influenza A virus-challenged mice (S3A and S3B Fig).

Nuclear expression of proliferation-associated protein Ki67 revealed that 80% of total CD8+ T cells were committed to cell cycle by day 4 post re-infection (S3C Fig). We further analyzed expression of other lung residency-associated surface markers such as CD49a and CXCR6 [5,12]. Consistent with results on CD103 and CD69 expression, lung resident (i.v.-) RSV-specific CD49a+ and CXCR6+ CD8+ T cells peaked at day 4 post re-infection (S3D and S3E Fig). Of note, the majority of lung CD69+ CD103+ CD8+ T cells co-expressed CD49a and CXCR6. These data show that lung CD8+ T cells expressing residency-associated markers expand during RSV re-infection. While expansion in the lung was evident, lung draining lymph nodes showed modest expansion of resident (i.v.CD45-) $M_{187-195}$ CD8+ T cells 4 days post RSV re-infection, but as expected, these cells lacked CD69 and CD103 co-expression (S3F Fig).

During secondary challenge in mice, RSV viral replication was rapidly controlled, being mostly cleared by day 4 post re-infection (Fig 2A). Secondary challenge also induced an innate IFN response (Fig 2B) although levels of IFN-α at day 1 post re-infection were lower than those detected during primary infection [32]. Similarly, mRNA expression of *Cxcl9* and *Cxcl10* in lung tissue peaked 1 day after re-challenge (Fig 2C and 2D). To evaluate the function of RSV-specific $T_{RM}$ cells, lung and airway cells were stimulated with RSV-specific $M_{187-195}$ peptide and intracellular IFN-γ and granzyme B (GzmB) expression in CD8+ T cells were assessed by flow cytometry (Fig 2E). CD103+ CD69+ CD8+ $T_{RM}$ cells produced IFN-γ and GzmB and these cells increased by day 4 post re-infection in the lung and airways (Fig 2F and 2G). Altogether these results characterize the CD8+ T cell response during RSV secondary infection and show that functional RSV-specific CD69+ CD103+ CD8+ $T_{RM}$ cells in the lung and airways go through an expansion between days 3–4 post re-infection.

## $T_{RM}$ cell expansion during RSV secondary infection is sustained independently of recruitment of circulatory T cells

$T_{RM}$ cells are defined by residence in peripheral tissues and lack of exchange with the recirculating pool in addition to the expression of certain surface markers [12,41]. To identify lung

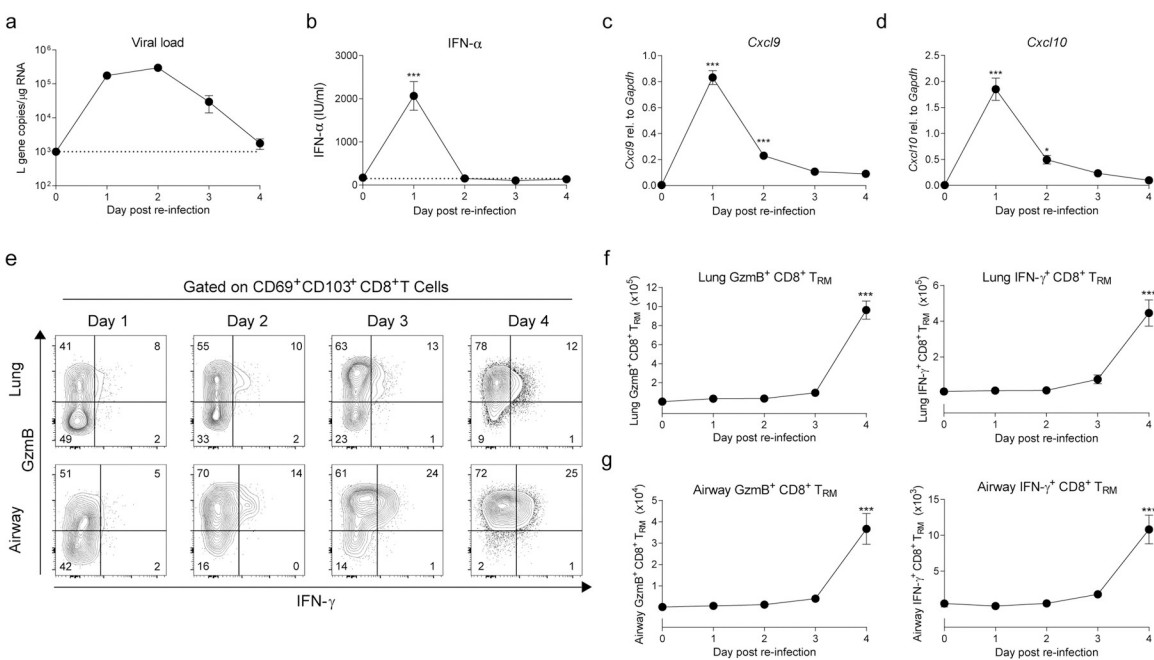

**Fig 2. Inflammatory mediators during RSV-mediated CD8+ T_RM cell re-activation.** Mice were RSV infected i.n. and three weeks p.i. mice were mock or re-challenged with RSV and lung and airways samples were analyzed at different time points (day 0 = mock re-infection) (**a**) Lung viral load determined by RT-qPCR quantification of RSV L gene expression in lung tissue. (**b**) IFN-α levels in BAL fluid analyzed by ELISA. mRNA expression of (**c**) *Cxcl9* and (**d**) *Cxcl10* was quantified by RT-qPCR in lung tissue. Lung and airway cells were stimulated with RSV M_{187-195} peptide and IFN-γ and GzmB production was detected by intracellular staining and quantified in CD8+ T_RM cells using flow cytometry. (**e**) Representative flow cytometry plots of IFN-γ and GzmB intracellular staining in CD8+ CD69+ CD103+ T_RM cells. Total number of IFN-γ and GzmB positive CD8+ CD69+ CD103+ T_RM cells in (**f**) lung tissue and (**g**) airways. Data are presented as the mean±SEM of 9–11 individual mice per time point, pooled from two independent experiment. Statistical significance of differences between day 0 (mock re-infection) and other time points was determined by one-way ANOVA with Tukey's post hoc test. * indicates differences between day 0 and days 1–4. * P ≤ 0.05, ** P ≤ 0.01, *** P ≤ 0.001.

resident CD8+ T cells during RSV secondary challenge, mice were treated with FTY720, a sphingosine-1 phosphate receptor down regulator that blocks T cell egress from lymphoid tissues and thereby sequesters recirculating T cells within secondary lymphoid organs [42]. Mice were given FTY720 (25 μg) daily from day -2 to day 3 post re-infection and at day 4 immune cell populations were analyzed in the lung tissue cell suspensions (Fig 3A). As before, mice were injected with BUV395 conjugated anti-CD45 antibody (CD45 i.v.) 10 min before euthanasia to identify cells in the lung vasculature that end up in the lung cell suspension (gating strategy S1C Fig). As expected, AMs showed very low CD45 i.v. staining, while most of the neutrophils were positive for CD45 i.v. staining, consistent with residence in the vasculature (S4A and S4B Fig). CD4+ T cells showed mixed composition of CD45 i.v.+ and i.v.−: naïve CD4+ T cells were positive for CD45 i.v. staining and their presence was abrogated by FTY720 treatment (S4C and S4D Fig) while effector CD4+ T cells were both CD45 i.v.+ and i.v.- and only partially lost upon FTY720 treatment, indicating mixed contribution from vasculature and lung parenchyma (S4E Fig). On the other hand, almost all CD8+ T cells in the lung were negative for CD45 i.v. staining and numbers of i.v. CD45- total CD8+ T cells were unaltered by FTY720 treatment (S4F–S4H Fig). In blood, FTY720 treatment diminished CD4+ and CD8+ T cell but not neutrophil numbers (S4I–S4K Fig). Notably, lung CD69+ CD103+ CD8+ T_RM cells expanded in FTY720 treated mice to the same extent as in the vehicle treated mice (Fig 3B and 3C). Interestingly, CD69+ CD103- T cells were also unaffected by FTY720 treatment suggesting that these cells are also T_RM cells even if they do not express CD103 (Fig 3D). FTY720

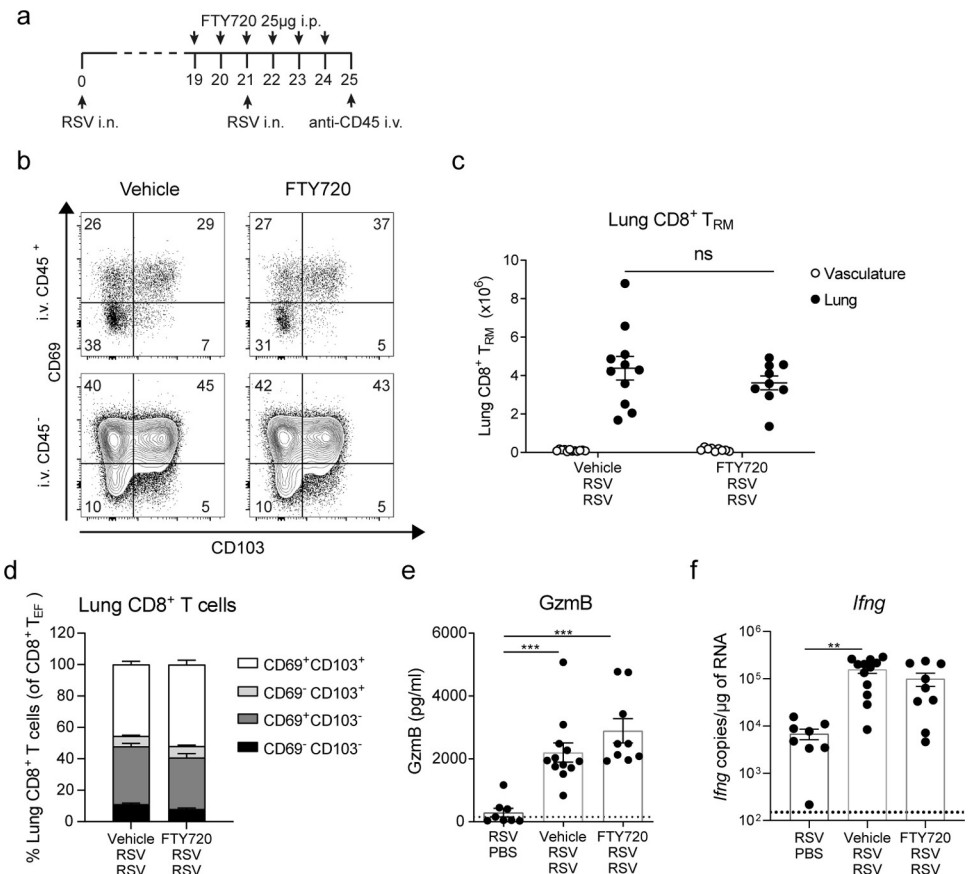

**Fig 3. Lung CD8+ T_RM cell expansion after RSV re-infection is sustained independently of circulatory T cells.** (**a**) Mice were infected with RSV i.n. and 3 weeks p.i. mice were re-challenged i.n. with RSV (RSV/RSV). Mice were treated with 25 μg FTY720 i.p. administered daily from day -2 prior to re-challenge until day 3 post re-challenge. Mice were given 2μg CD45-BUV395 i.v. 10 min before euthanasia to distinguish cells in the vasculature of the lung (i.v. CD45+) from resident lung cells (i.v. CD45-). (**b**) Representative flow cytometry plots of CD69 and CD103 expression on intravascular stained or unstained CD8+ T_EF cells (CD44+ CD62L-). (**c**) Total number of CD8+ T_RM cells in the vasculature of the lung and in the lung parenchyma. (**d**) Stacked analysis of the subpopulations of CD8+ T_EF cells based on expression of CD103 and/or CD69. (**e**) GzmB levels in BAL fluid were determined by ELISA and (**f**) *Ifng* mRNA expression in lung tissue was assessed by RT-qPCR. Data are presented as the mean±SEM of 7 mock re-infected (RSV/PBS), 12 vehicle-treated and 9 FTY720-treated RSV/RSV individual mice pooled from two independent experiment. Statistical significance of differences between groups was determined by one-way ANOVA with Tukey's post hoc test. * P ≤ 0.05, ** P ≤ 0.01, *** P ≤ 0.001.

treatment did not affect GzmB levels in BAL fluid (Fig 3E) nor *Ifng* mRNA expression in the lungs (Fig 3F), implying that lung resident cells are primarily responsible for the production of these mediators at day 4 post re-infection. Altogether, these observations reveal that CD8+ T cell short-memory responses are mainly mediated by T_RM cells and that expansion of CD8+ T_RM cells is sustained independently of recruitment of recirculating CD8+ T cells.

## MAVS and MyD88/TRIF signaling are necessary for T_RM cell expansion during RSV secondary infection

To assess the contribution of PRR signaling to the CD8+ T_RM cell responses, *Mavs*-/- and *MyD88/Trif*-/- mice were infected with RSV. During primary infection *Mavs*-/- mice lost more weight than wt and *MyD88/Trif*-/- mice (S5A Fig and [32]). After 25 days post-infection, all mice showed detectable lung CD8+ T_RM cells, although *Mavs*-/- mice had lower frequency and

numbers (Fig 4A). Wt, *Mavs*⁻/⁻ and *Myd88/Trif*⁻/⁻ mice showed the same proportions of RSV-specific CD8+ $T_{RM}$ cells (S5B Fig). After RSV re-infection viral load was detected at day 2 (peak of viral load, Fig 2) and 4 in wt, *Mavs*⁻/⁻ and *Myd88/Trif*⁻/⁻ mice (S6A Fig). There was no difference between the groups at day 2 although, wt mice displayed more rapid clearing of the virus at day 4 post re-infection (S6A Fig). To better characterize the immune response during re-infection, immune cell infiltration and chemokines were assessed in wt, *Mavs*⁻/⁻, and *Myd88/Trif*⁻/⁻ mice (gating strategy in [32]). Resembling what was described during primary infection, *Myd88/Trif*⁻/⁻ mice showed less recruitment of neutrophils to the lung and airways, while *Mavs*⁻/⁻ mice showed lower numbers of inflammatory monocytes in the lung (S6B-S6G Fig). Chemokine induction was also measured, and all groups showed an induction of *Cxcl10*, *Ccl2* and *Cxcl1* expression in the lung after RSV re-infection (S6H-S6J Fig). Although the innate response was similar between the groups, both *Mavs*⁻/⁻ and *Myd88/Trif*⁻/⁻ mice had reduced frequencies and numbers of CD69+ CD103+ $T_{RM}$ cells in the lung and the airways (Fig 4B–4D). Compared with numbers prior to RSV re-infection, wt mice showed an 18.5 fold increase in the numbers of $T_{RM}$ cells, while *Mavs*⁻/⁻ mice a 10.5 fold and *Myd88/Trif*⁻/⁻ mice a 10.8 fold increase (Fig 4A and 4C). RSV-specific CD8+ $T_{RM}$ cells were detected using tetramers loaded with RSV $M_{187-195}$ epitope (gating strategy S1D Fig). $M_{187-195}$-specific CD8+ CD69+CD103+ $T_{RM}$ cells were reduced in the lung of *Mavs*⁻/⁻ mice (Fig 4E). *Myd88/Trif*⁻/⁻ mice had lower frequency of RSV-specific $T_{RM}$ cells compared to wt mice (Fig 4E) but similar numbers due to increased levels of total CD8+ T cells (Figs 4E and S5D). A closer look at the RSV-specific CD8+ T cells revealed that the loss of $T_{RM}$ cells in *Mavs*⁻/⁻ mice was restricted to the CD69+ CD103+ sub-population and not as prominent for CD69+ CD103⁻ $T_{RM}$ cells (Fig 4F). To determine if this was solely due to a defect on surface expression of CD103, surface expression of CD49a and CXCR6 was analyzed on CD8+ $M_{187-195}$-specific T cells independently of CD103 or CD69. Consistently, CD49a+ and CXCR6+ lung RSV-specific CD8+ T cells were less abundant in *Mavs*⁻/⁻ and *Myd88/Trif*⁻/⁻ compared to wt mice (Fig 4G and 4H).

To validate our findings, dimensionality reduction analysis was conducted on $M_{187-195}$-specific CD8+ T cells in wt, *Mavs*⁻/⁻ and *Myd88/Trif*⁻/⁻ mice using Uniform Manifold Approximation and Projection (UMAP) algorithm and FlowSOM software to identify major clusters in the samples. The analysis confirmed that the predominant phenotype of RSV-specific CD8+ T cell is strikingly different between wt, *Mavs*⁻/⁻ and *Myd88/Trif*⁻/⁻ mice (Fig 4I). The largest cluster of RSV-specific CD8+ T cells co-expressed CD69, CD103, CD49a and CXCR6 in wt mice (population 5). In contrast, *Mavs*⁻/⁻ and *Myd88/Trif*⁻/⁻ mice showed a large cluster of CD103⁻ CD69+ CD49a+ CXCR6+ CD8+ T cells (population 2). Data obtained using dimensionality reduction resembles data obtained by manual gating: while wt mice had a predominant CD69+ CD103+ population in the RSV-specific CD8+ T cell pool, absence of PRR signaling reduces RSV-specific CD69+ CD103+ $T_{RM}$ cells during RSV re-infection.

## $T_{RM}$ cell functionality is impaired in MAVS deficient mice during RSV secondary infection

Next, we focused on the functional response of $T_{RM}$ cells in *Mavs*⁻/⁻ and *Myd88/Trif*⁻/⁻ mice. Day 4 post re-infection, lung and airway cells were stimulated *ex vivo* with RSV $M_{187-195}$ peptide and intracellular presence of GzmB and IFN-γ was assessed by flow cytometry (Fig 5A). Almost 90% of CD8+ $T_{RM}$ cells in the lung and airways of wt and *Myd88/Trif*⁻/⁻ mice were positive for GzmB and 35–60% for IFN-γ (Fig 5B and 5C). In contrast, only 70% of the lung and airway $T_{RM}$ cells in *Mavs*⁻/⁻ mice were able to produce GzmB and 20–40% IFN-γ after $M_{187-195}$ peptide stimulation (Fig 5B and 5C). In combination with the decreased number of CD69+ CD103+ $T_{RM}$ cells in MAVS deficient mice, this markedly reduced the total number of $T_{RM}$

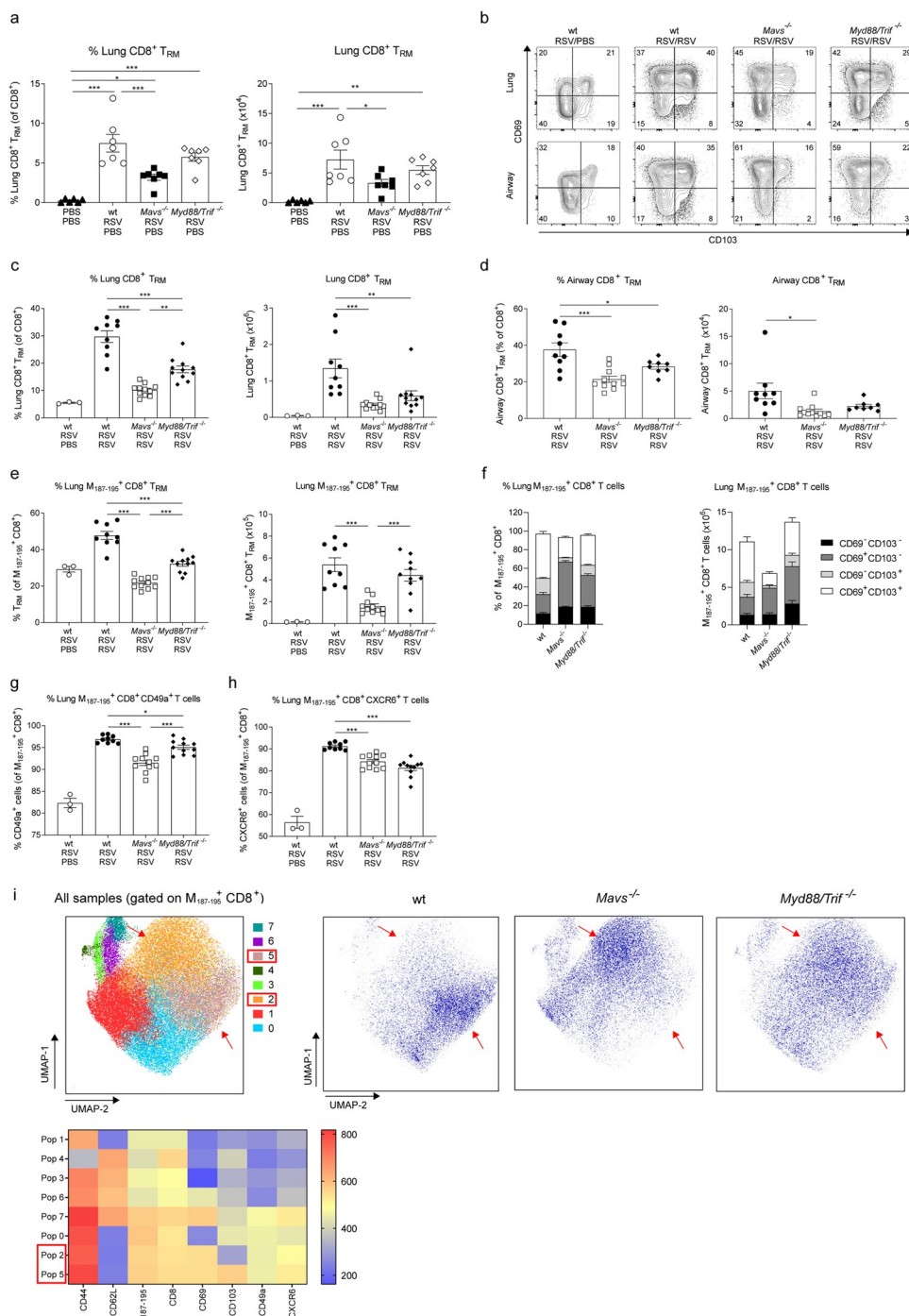

**Fig 4. PRR signaling is required for T$_{RM}$ cell expansion during RSV secondary challenge.** Wt, MAVS (*Mavs$^{-/-}$*) and MyD88/TRIF deficient (*Myd88/Trif $^{-/-}$*) mice were infected with RSV and 25 days later lung cells were recovered and compared with mock infected (PBS/PBS) wt mice. Percentage and numbers of CD8+ T$_{RM}$ were analyzed by flow cytometry (**a**). Data are represented as mean±SEM of 7 mice per group pooled from two independent experiments. Wt, *Mavs$^{-/-}$* and *Myd88/Trif $^{-/-}$* mice were infected with RSV and three weeks later re-infected with RSV. Four days after re-infection mice were euthanized and airway and lung cells were recovered and the CD8+ T$_{RM}$ cell populations analyzed. (**b**) Representative flow cytometry plots showing CD69 and CD103 expression on lung and airway CD8+ T$_{EF}$ cells. Percentage of CD103+ CD69+ T$_{RM}$ cells in CD8+ T$_{EF}$ cells and total number of (**c**) lung and (**d**) airway CD69+ CD103+ T$_{RM}$ cells in wt, *Mavs$^{-/-}$* and *Myd88/Trif $^{-/-}$* re-infected mice. Lung cells were stained with RSV M$_{187-195}$ containing tetramers to identify RSV-specific CD8+ T cells. (**e**) Percentage of M$_{187-195}$ specific CD69+ CD103+ T$_{RM}$ cells of M$_{187-195}$ specific CD8+ T cells and total number of M$_{187-195}$ specific CD69+ CD103+ T$_{RM}$ cells. (**f**) Stacked

analysis of the percentage and number of M$_{187-195}$ specific CD8$^+$ T$_{EF}$ cells based on CD103 and CD69 expression. Expression of (**g**) CD49a and (**h**) CXCR6 on M$_{187-195}$ specific CD8$^+$ T cells was quantified as a percentage of M$_{187-195}$ specific CD8$^+$ T cells. (**i**) M$_{187-195}$ specific CD8$^+$ T cells from RSV re-infected wt, *Mavs$^{-/-}$* and *Myd88/Trif$^{-/-}$* mice were concatenated and analyzed using unsupervised dimensionality reduction software (UMAP). Clusters were automatically identified using FlowSOM and density plots of concatenated (All samples) with identified populations and wt, *Mavs$^{-/-}$* and *Myd88/Trif$^{-/-}$* are shown. Heatmaps with the marker expression of each of the 8 populations identified are shown. Populations with mayor differences between groups are indicated with red arrows and squares in the legend. Data are presented as the mean±SEM of 9 re-infected wt, 11 *Mavs$^{-/-}$* and 11 *Myd88/Trif$^{-/-}$* mice from two independent experiment. Statistical significance of differences between groups was determined by one-way ANOVA with Tukey's post hoc test. * P $\leq$ 0.05, ** P $\leq$ 0.01, *** P $\leq$ 0.001.

cells displaying effector function in response to RSV secondary challenge in *Mavs$^{-/-}$* mice (S5E and S5F Fig). Importantly, decreased functionality was restricted to the CD69$^+$ CD103$^+$ T$_{RM}$ cells in *Mavs$^{-/-}$* mice, as CD8$^+$ CD69$^+$ CD103$^-$ T cells were capable of responding in a similar fashion to wt or *Myd88/Trif$^{-/-}$* mice (Fig 5D). Of note, the defect in T$_{RM}$ cell effector functions was also evident at the level of the overall lung response to re-challenge as *Mavs$^{-/-}$* mice had decreased GzmB levels in BAL and *Ifng* expression in lung tissue (Fig 5E and 5F). In contrast, *Myd88/Trif$^{-/-}$* mice contained *Ifng* mRNA in lung tissue and even displayed increased levels of GzmB in BAL fluid compared to wt mice (Fig 5E and 5F). T$_{RM}$ cell numbers were similar in the absence or presence of *ex vivo* peptide stimulation, ruling out possible alteration of T$_{RM}$ cell numbers by *ex vivo* peptide stimulation, (S5G Fig).

To validate our observations, UMAP analysis was conducted on RSV peptide-stimulated CD8$^+$ T cells. This confirmed striking differences in the phenotypic composition of activated CD8$^+$ T cell in *Mavs$^{-/-}$* mice compared to wt and *Myd88/Trif$^{-/-}$* mice (Fig 5G, density plots). In addition, cluster identification analysis using FlowSOM organized the data according to parameter similarity into 8 clusters (Populations 0–7) (Fig 5G, heatmap). The data further showed that GzmB and IFN-γ producing CD103$^+$ CD69$^+$ T$_{RM}$ cells were very underrepresented in *Mavs$^{-/-}$* mice (populations 1 and 3), although displaying phenotypically similar CD69$^+$ CD103$^-$ responding CD8$^+$ T cells (Pop 0 and 2) (Figs 5G and S5H). Altogether, our results demonstrate that MAVS is necessary to induce diverse and functional RSV-specific T$_{RM}$ cells during RSV re-infection, while MyD88/TRIF signaling is dispensable.

## Type I IFN responses during primary infection are necessary for T$_{RM}$ cell generation during short-term memory responses

MAVS-dependent type I IFNs drive inflammation after RSV infection [32,43]. Lung inflammation absent in *Mavs$^{-/-}$* mice can be partially restored when animals are treated with IFN-α i. n. early during primary RSV infection [32]. We reasoned that this early inflammatory response could affect the subsequent generation of a memory response. To test this, *Mavs$^{-/-}$* mice were treated with two doses of recombinant IFN-α (500 ng/mice) at 6h and 18h post-primary infection (Fig 6A). IFN-α treated *Mavs$^{-/-}$* mice displayed increased weight loss during primary RSV infection compared to wt and untreated *Mavs$^{-/-}$* mice (Fig 6B). Interestingly, when IFN-α treated *Mavs$^{-/-}$* mice were re-infected 3 weeks after primary infection, CD69$^+$ CD103$^+$ CD8$^+$ T$_{RM}$ cell expansion was recovered in the lung and airways (Fig 6C and 6D). This recovery was also patent at the level of the number and frequency of RSV-specific T$_{RM}$ cells (S5I Fig). The viral load at day 2 and 4 post re-infection was similar in the IFN-α treated *Mavs$^{-/-}$* mice compared to the other groups (S6A Fig). However, GzmB levels in BAL and levels of *Ifng* mRNA in total lung remained significantly lower in *Mavs$^{-/-}$* groups compared to wt mice (Fig 6E and 6F) irrespective of IFN-α treatment. Furthermore, even though IFN-α treatment normalized the number of T$_{RM}$ cells in *Mavs$^{-/-}$* mice, cell functionality remained impaired (Fig 6G) as the frequency of GzmB$^+$ or IFN-γ$^+$ CD69$^+$ CD103$^+$ T$_{RM}$ cells was decreased in the lungs and BAL

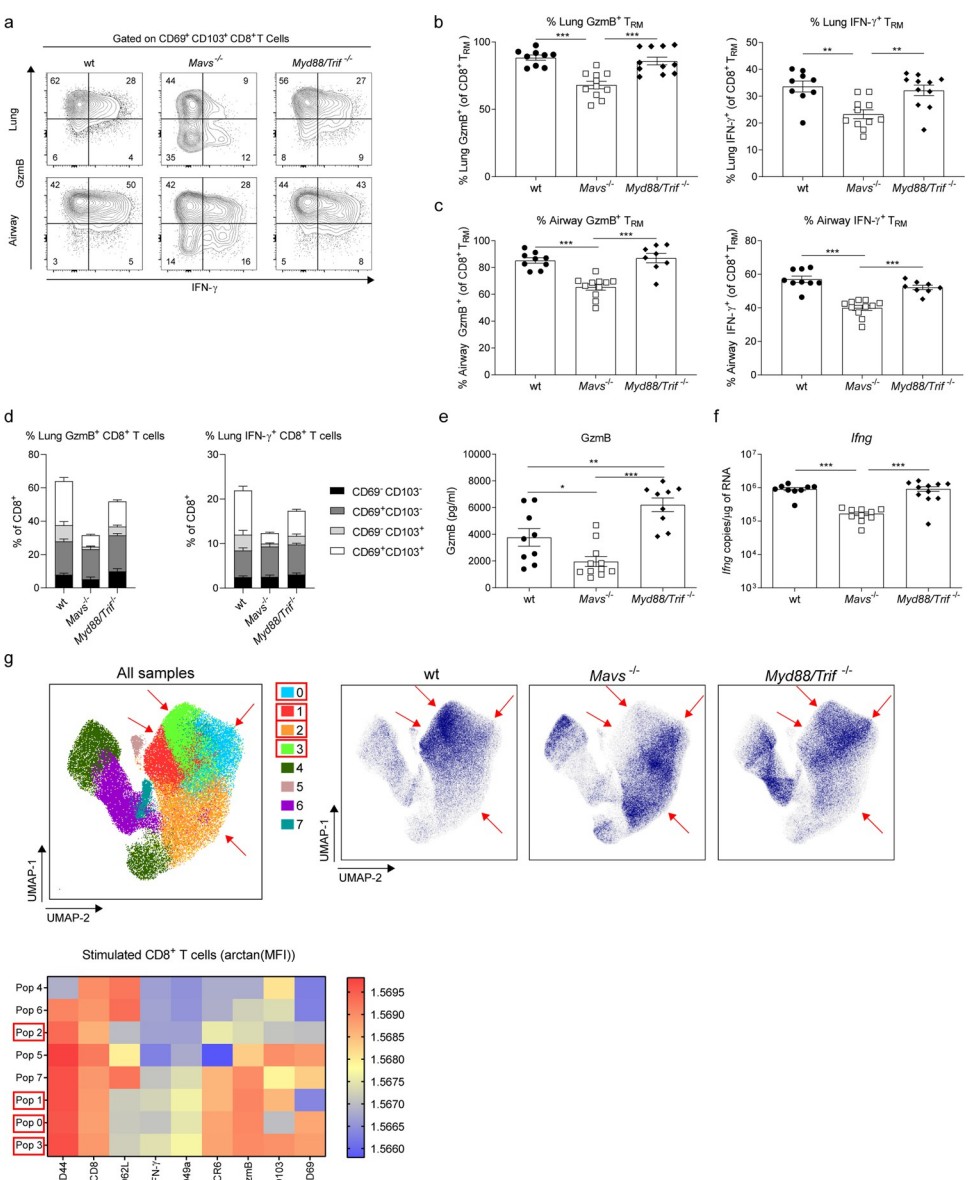

**Fig 5. MAVS deficient mice show functionally impaired T$_{RM}$ cells during RSV re-infection.** RSV re-infected wt, *Mavs*$^{-/-}$ and *Myd88/Trif*$^{-/-}$ mice were euthanized after four days and airway and lung cells stimulated with RSV M$_{187-195}$ peptide and CD8⁺ T cell intracellular production of GzmB and IFN-γ was determined by flow cytometry. (**a**) Representative flow cytometry plots of GzmB and IFN-γ producing CD69⁺ CD103⁺ T$_{RM}$ cells. Percentage of GzmB⁺ and IFN-γ⁺ CD103⁺ CD69⁺ T$_{RM}$ cells were quantified as a percentage of total CD103⁺ CD69⁺ T$_{RM}$ cells in the (**b**) lungs and (**c**) airways. (**d**) Stacked analysis of percentage of GzmB⁺ or IFN-γ⁺ T$_{EM}$ cells subsets discriminated by CD69 and CD103 surface expression. (**e**) GzmB was quantified by ELISA in BAL fluid. (**f**) *Ifng* mRNA detected in lung tissue using RT-qPCR. (**g**) Lung cells were stimulated with RSV M$_{187-195}$ peptide and stained for flow cytometry analysis. Total CD8⁺ T cells were downsampled and concatenated and this file was submitted to unsupervised dimensionality reduction using UMAP software. Clusters were automatically identified using FlowSOM and density plots for concatenated (All samples) including identified populations for each genotype is shown. Heatmap with the marker expression of each of the 8 populations identified is also depicted. Populations that show mayor differences are pointed with red arrows and squares in the legend. Data are presented as the mean±SEM of 9 re-infected wt, 11 re-infected *Mavs*$^{-/-}$ and 11 re-infected *Myd88/Trif*$^{-/-}$ mice pooled from two independent experiment. Statistical significance of differences between groups was determined by one-way ANOVA with Tukey's post hoc test. * P ≤ 0.05, ** P ≤ 0.01, *** P ≤ 0.001.

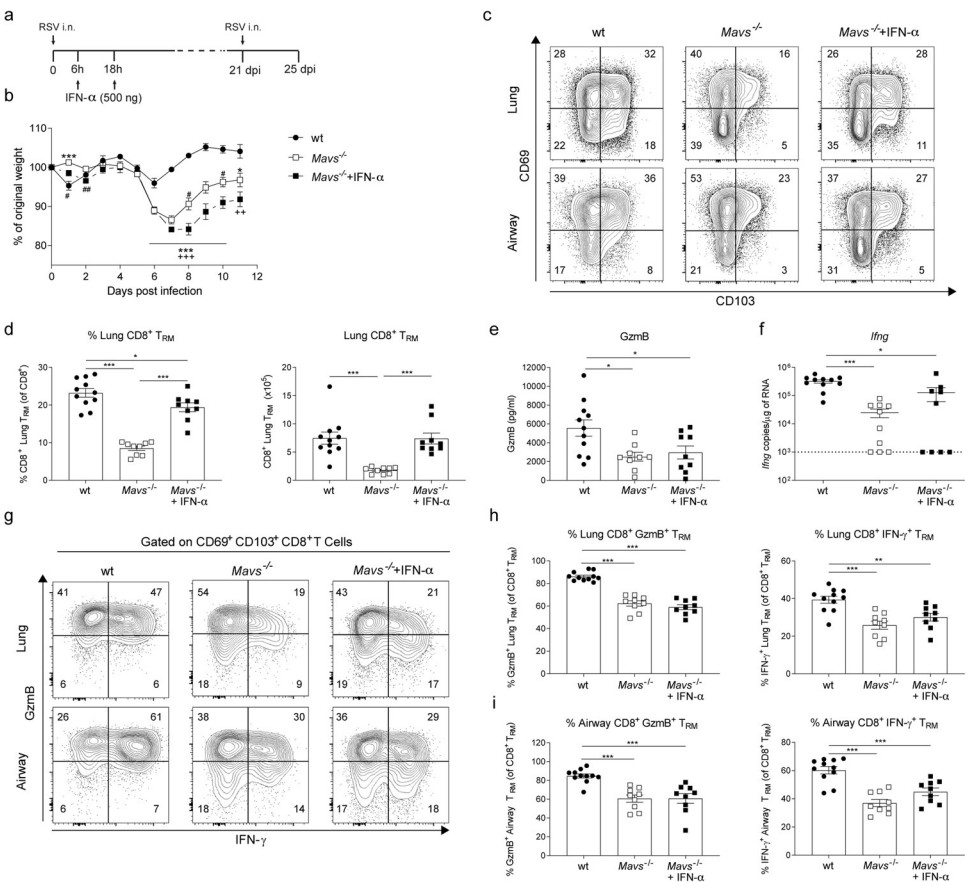

**Fig 6. Early type I IFNs during primary infection is needed for generation of CD8+ T$_{RM}$ cells detected during RSV re-infection.** (**a**) *Mavs*$^{-/-}$ mice were treated i.n. at 6h and 18h during primary RSV infection with 500ng of recombinant IFN-α. (**b**) Body weight was monitored throughout primary infection in wt, *Mavs*$^{-/-}$ and rIFN-α treated *Mavs*$^{-/-}$ mice and percentage of original weight was quantified. Twenty-one days later mice were re-infected with RSV and CD8+ T$_{RM}$ cell responses were analyzed by flow cytometry 4 days post re-infection. (**c**) Representative flow cytometry plots showing CD69 and CD103 surface expression on CD8+ T$_{EF}$ cells. (**d**) Percentage of lung CD103+ CD69+ T$_{RM}$ cells in total CD8+ T cells and absolute numbers in lung were quantified. (**e**) GzmB levels in BAL fluid was quantified by ELISA. (**f**) *Ifng* mRNA expression in lung tissue was quantified by RT-qPCR. Lung and BAL cells were stimulated *ex vivo* with RSV M$_{187-195}$ peptide and intracellular GzmB and IFN-γ production was determined in CD103+ CD69+ T$_{RM}$ cells by flow cytometry. (**g**) Representative flow cytometry plots showing lung and airway GzmB and IFN-γ intracellular expression in CD103+ CD69+ T$_{RM}$ cells. Proportion of lung (**h**) and airway (**i**) GzmB+ and IFN-γ+ CD103+ CD69+ T$_{RM}$ cells as a percentage of CD103+ CD69+ T$_{RM}$ cells. Data are presented as the mean±SEM of 11 re-infected wt, 9 re-infected *Mavs*$^{-/-}$ and 9 rIFN-α treated re-infected *Mavs*$^{-/-}$ mice pooled from two independent experiment. Statistical significance of differences between groups was determined by one-way ANOVA with Tukey's post hoc test. In panel b * represent differences between wt and *Mavs*$^{-/-}$; $ differences between *Mavs*$^{-/-}$ and *Mavs*$^{-/-}$+IFN-α, and + differences between wt and *Mavs*$^{-/-}$+IFN-α. * P ≤ 0.05, ** P ≤ 0.01, *** P ≤ 0.001.

irrespective of IFN-α treatment (Fig 6H and 6I). To note, the total numbers of GzmB+ and IFN-γ+ T$_{RM}$ cells in lung and airways were increased in IFN-α treated *Mavs*$^{-/-}$ mice, compared to untreated *Mavs*$^{-/-}$ mice, as these had more total T$_{RM}$ cells (S5J and S5K Fig). Overall, these observations indicate distinct MAVS-dependent mechanisms regulating T$_{RM}$ cell expansion and T$_{RM}$ cell functionality during RSV secondary infection.

## Discussion

CD8+ T$_{RM}$ cells act as a frontline defense during re-infections mainly due to their functional readiness and strategic location in the tissues. The presence of antigen-specific T$_{RM}$ cells in the

lungs is associated with lower viral load during re-infections [15,17,19,44–48]. Here, we studied the nature of CD8$^+$ T$_{RM}$ cell responses during RSV re-infection using a well-established murine RSV infection model [32]. We characterized the dynamics and functional capacity of lung CD8$^+$ T$_{RM}$ cells, showing an expansion at day 4 post RSV re-infection. FTY720 treatment during re-infection, revealed that both CD69$^+$ and CD103$^+$ CD69$^+$ CD8$^+$ T$_{RM}$ cells are expanded from pre-existing lung T$_{RM}$ cells with minor contribution from circulatory T cells. Moreover, PRRs signaling deficient, *Mavs$^{-/-}$* and *Myd88/Trif$^{-/-}$*, mice showed reduced expansion of the CD69$^+$ CD103$^+$ CD8$^+$ T$_{RM}$ cells during secondary RSV challenge. However, only *Mavs$^{-/-}$* but not *Myd88/Trif$^{-/-}$* mice showed impaired T$_{RM}$ cell functionality with lower frequency of cells producing IFN-γ and GzmB after RSV-specific peptide stimulation. Strikingly, T$_{RM}$ cell expansion but not T$_{RM}$ cell functional impairment in *Mavs$^{-/-}$* mice was restored after IFN-α treatment, possibly revealing a dual role for MAVS signaling via IFN-dependent and -independent pathways to regulate the T$_{RM}$ cell pool during RSV re-infection.

Our results show that RSV infection results in the generation of lung CD8$^+$ T$_{RM}$ cells, which expand after secondary challenge. The direct effect of T$_{RM}$ cells on clearing RSV is difficult to assess in the mouse model as, in contrast to humans, mice are capable of producing and maintaining RSV-specific antibody responses to rapidly inhibit infection [39]. As a consequence, viral load at the peak of RSV re-infection is 1000-fold lower than in primary infection [32]. During RSV re-infection, despite major differences in T$_{RM}$ cell numbers and function in wt mice compared with *Mavs$^{-/-}$*, we found no differences in viral load at day 2 post re-infection although, in these experiments, wt mice showed slightly faster viral clearance than *Mavs$^{-/-}$* mice at day 4 post re-infection. However, low viral load may contribute to some experimental variability as we had previously found no difference in viral load day 4 post re-infection [39]. Arguing for a role of T$_{RM}$ cell in RSV control, other groups have shown that, in the absence of antibodies, adoptive transfer of airway CD8$^+$ T cells is enough to control disease development [18]. Further, RSV infected mice re-infected with recombinant IAV expressing RSV T cell epitopes and kept under FTY720 treatment are partially able to control viral load [19]. Finally, T$_{RM}$ cell generation is reduced during infancy in mice and these mice fail to control influenza virus replication during heterosubtypic infection compared to adult mice [23]. These observations in mice draws a parallel with human RSV infection where children are highly susceptible to symptomatic infection compared to adults, consistent with the notion that T$_{RM}$ cell responses play a role in viral control during re-infection.

We show that CD103$^-$ CD69$^+$ and CD103$^+$ CD69$^+$ CD8$^+$ T cells are truly lung resident, and that recruitment from the blood is minimal during re-infection by using FTY720 treatment complemented with intravascular CD45 staining. In contrast, during influenza virus heterosubtypic infection, CD69$^+$ CD103$^-$ T$_{RM}$ cells predominate over CD69$^+$ CD103$^+$ T$_{RM}$ cells and some contribution from the circulation is also observed [16]. Consistently with our observations, intranasal RSV vaccination with an MCMV vector generated both CD69$^+$ CD103$^+$ and CD69$^+$ CD103$^-$ CD8$^+$ T$_{RM}$ cells at similar levels [49]. Altogether, these observations suggest that surface expression of CD103 on CD8$^+$ T$_{RM}$ cells depends on the environment generated by the different respiratory pathogens and supports the notion that CD8$^+$ T$_{RM}$ cells are heterogeneous and context-dependent even in the same mucosal tissue [12].

We have previously reported that alveolar macrophages in MAVS deficient mice are unable to produce type I IFN during primary RSV infection [32]. This affects the early dynamics of the innate immune response (i.e. neutrophil activation, monocyte recruitment, IFN-γ, TNF-α, IL-6, IL-1β and CXCL1 production) and causes delayed viral control and heightened weight loss [32,33]. Interestingly, later events were comparable between MAVS deficient and wt mice, and there were no major differences in immune cell (neutrophils, inflammatory monocytes, CD4$^+$ and CD8$^+$ T cells) recruitment to the lungs from day 2–9 post primary infection [39].

Treatment of MAVS deficient mice with type I IFNs recovered early monocyte recruitment to the lungs, neutrophil activation and IFN-γ, TNF-α, IL-6 production to levels comparable to wt mice [32,33]. Here we report that MAVS deficient mice also regulate CD8+ T cell memory generation and proliferative capacity in a type I IFN dependent fashion.

TLR3 and 7 engagement play a role during RSV infection, although not as marked as signaling via RIG-I-like receptors (RLRs) [50]. Lack of TLR signaling via adaptor proteins MyD88 and TRIF in mice does not affect weight loss, morbidity or viral load during RSV primary infection (S4A Fig; [33,34]). However, the TLR pathways regulate specific immune events during RSV infection such as CXCL1-mediated neutrophil recruitment to the lungs [33]. Although MyD88 and TRIF signaling is not required for mounting functional adaptive T cell responses [34,40], we show that it is required for proper expansion of CD8+ $T_{RM}$ cells during RSV secondary challenge. As type I IFNs are produced at almost wt levels in *Myd88/Trif* $^{-/-}$ mice [33], impairment of CD8+ $T_{RM}$ cell generation in *Mavs* $^{-/-}$ and *Myd88/Trif* $^{-/-}$ mice must be explained by different mechanisms. In other viral infections (herpes simplex type I virus, hepatitis B virus, vaccinia virus, lymphocytic choriomeningitis virus, and IAV) TLR signaling is necessary for CD8+ T cell responses, acting either directly on CD8+ T cells or by indirect mechanisms affecting T cell priming [51–56]. In contrast, in the RSV model, $T_{RM}$ cell production of IFN-γ and GzmB was not affected by *Myd88/Trif* genetic ablation, showing that this is not a requirement for RSV-specific CD8+ $T_{RM}$ cell function.

It is interesting to consider how the IFN-α treatment during primary infection of *Mavs* $^{-/-}$ mice might result in increased $T_{RM}$ cell expansion. As modulation of conventional dendritic cell function (i.e. migration, maturation, antigen processing and cross-presentation) can be regulated by type I IFNs [57,58] this could be a possible mechanism that could explain our data. Type I IFNs can also directly act on T cells during priming thus determining T cell memory and residency programs and it has been shown that CD8+ T cells cultured with type I IFNs upregulate CD103 [37,38].

CD8+ $T_{RM}$ cell functionality is not recovered in IFN-α treated *Mavs* $^{-/-}$ mice suggesting a differential regulation of the activation process independently of the expansion of CD8+ $T_{RM}$ cells. How MAVS deficiency affects the functional response of $T_{RM}$ cells independently of type I IFNs still needs further investigation. However, it is interesting to note that only CD103+ CD69+ CD8+ $T_{RM}$ cells but not CD69+ CD103- CD8+ $T_{RM}$ cells are affected by MAVS deficiency, showing that specific subsets may have different requirements for activation. Consistent with our data, Kohlmeier et al. have shown that type I IFNs during murine influenza virus re-infection enhances effector functions of memory CD8+ T cells recruited to the lungs [59]. It is therefore possible that there are CD8+ $T_{RM}$ cell intrinsic mechanisms linking MAVS signaling or type I IFN signaling with function. Engagement of IFNAR in CD8+ T cells has been reported to regulate function and STAT-1 signaling inhibits T cell activation [60]. Whether similar mechanisms govern CD8+ $T_{RM}$ cell response is still unclear. In addition, Kaech and colleagues have shown that $T_{RM}$ cell re-activation in the lungs is affected by signals from different antigen presenting cells [61]. It is likely that type I IFNs, present during re-infection, may indirectly affect CD8+ T cell activation via modulation of antigen presentation. Further investigation should tackle the extrinsic or intrinsic mechanism that drives MAVS-dependent CD8+ $T_{RM}$ cell activation during respiratory virus re-infection.

The importance of eliciting long-lasting tissue resident memory response after natural or artificial immunization has been suggested by some as a means to avoid future infections [12,15,41]. Our results on RSV infection and those obtained from influenza virus infection studies (by Kohlmeier et al.), suggest type I IFNs as appealing factors to consider for design of mucosal vaccines to induce lung resident T cell responses [59]. Moreover, a particularity of lung CD8+ $T_{RM}$ cells is that they show short lifespan compared with those present in skin and

other mucosal tissues [48,62]. After murine RSV infection, lung $T_{RM}$ cells gradually wane by day 149 post-infection [19]. Whether type I IFNs responses affect longevity of cellular response in the lung has not been analyzed in this report and should be investigated in RSV and other respiratory infections. Interestingly, selective induction of IFN-induced transmembrane protein IFITM3 increases the lifespan of lung CD8+ $T_{RM}$ cells after IAV infection in mice [63]. The longevity of the $T_{RM}$ cells will be a key consideration to study after SARS-CoV-2 infection. Tissue resident memory T cell responses have not been investigated yet in patients surviving SARS-CoV-2 infection and it will be important to assess the relationship between type I IFNs and CD8+ $T_{RM}$ cell responses in the lung during SARS-CoV-2 infection. In addition, Bastard et al., have described the presence of type I IFN auto-antibodies only in patients that develop severe COVID-19[64]. Therefore, it will be important to address if the presence of autoantibodies against type I IFNs might affect subsequent generation of CD8+ $T_{RM}$ cells. Furthermore, Zhang et al., demonstrated that loss-of-function variants in genes that govern TLR3 and IRF-7 mediated type I IFNs response, are overrepresented in patients with life-threatening COVID-19 compared to milder manifestations of the disease [65]. This resembles the observation that links polymorphisms in genes that control the IFN system with RSV disease severity and the fact that children with severe RSV infection show impaired type I IFN responses [66,67]. Again, characterizing lung CD8+ $T_{RM}$ cells in patients with these variants will be important to comprehend the relation between type I IFNs and lung resident cellular responses in these high-risk populations. Our contributions will help to harness CD8+ $T_{RM}$ cell maintenance in the lung as we have established a link between innate immunity, especially type I IFNs, and expansion and function of CD8+ $T_{RM}$ cells during RSV re-infection. This is important for our understanding of tissue-specific memory responses to respiratory viruses and should be considered for vaccine design against elusive respiratory infections.

## Materials and methods

### Ethics statement

All animal experiments were reviewed and approved by Animal Welfare and Ethical Review Board (AWERB) within Imperial College London and approved by the UK Home Office in accordance with the Animals (Scientific Procedures) Act 1986 (PPL P3AFFF0DD).

### Mice

C57BL/6 mice were obtained from Charles River (UK). In experiments involving genetically modified animals, wt mice, mice deficient in MAVS (*Mavs*$^{-/-}$) and MyD88/TRIF (*Myd88/Trif*$^{-/-}$) were obtained from S. Akira (World Premier International Immunology Frontier Research Center, Osaka University, Osaka, Japan, were used [68,69]. These strains were *Ifna6*$^{gfp/+}$ but since *Ifna6* expression was not a primary readout the mice are designated as wildtype (wt), *Mavs*$^{-/-}$ and *Myd88/Trif*$^{-/-}$ mice. All animals were bred and maintained in specific pathogen-free conditions. The mice were gender- and age-matched (7–12 weeks) in each experiment.

### Virus and infection

Plaque-purified human RSV A2 (originally from ATCC) was grown in HEp-2 cells with DMEM supplemented with 2% fetal calf serum (FCS) and 2mM L-glutamine. For infections animals were transiently anaesthetized with Isofluran and infected intranasally (i.n.) with 7.5x10$^5$ FFU of RSV in 100μl. Secondary infection was performed with 8-10x10$^5$ FFU in 100μl at day 21–24 post-primary infection. 250 PFU/mouse of influenza virus (X31; obtained from

John McCauley, The Francis Crick Institute, UK) were given i.n. in 100μl as already described for RSV.

### FTY720 and IFN-α treatment

Six doses of 25μg of FTY720 (Enzo Life Sciences, UK) in 250μl were injected i.p. at days -2, -1, 0, 1, 2, 3 relative to secondary RSV infection. At day 4, animals were culled 10 min after intravenous injection with 2 μg (in 200 μl) of anti-CD45 BUV395 (30-F11, BD). To check staining levels blood samples were withdrawn postmortem from the femoral vein, cells were isolated and assessed by flow cytometry.

At 6h and 18h after primary infection mice were treated i.n. after light anesthesia with 500 ng of recombinant IFN-α (Miltenyi Biotech, UK) in 100μl.

### Isolation of cells from airway (BAL) and lung

At day 4 post-secondary challenge (day 25–32 post-primary infection) mice were culled. Tracheae were exposed and bronchoalveolar lavage (BAL) was performed by flushing the lungs 3 times with 1 ml phosphate-buffered saline (PBS) containing 0.5 mM EDTA (Life Technology, Paisley, UK). The fluid obtained was centrifuged at 3,500 x $g$ for 5 min; supernatants were stored at -80°C for cytokine detection and cellular pellets were treated with ACK to remove red blood cells and used for cellular staining. Mice were perfused with 10ml of PBS and lungs were excised. Middle-right lobe was snap frozen for RNA purification, and the remaining 4 lobes were collected into C-Tubes (Milteny Biotech, Surrey, UK) containing complete DMEM (cDMEM; supplemented with 10% FCS, 2mM L-glutamine, 100 U/ml penicillin and 100 μg/ml streptomycin), 1 mg/ml Collagenase D (Roche, UK) and 30 μg/ml DNase I (Sigma Aldridge, Dorset UK) and processed with gentleMACS dissociator according to manufacturer's protocol. Lobes were digested for 45–60 min in at 37°C and further processed in the gentleMACS dissociator. Red blood cells in the homogenates were lysed using ACK buffer and then suspensions were washed, centrifuged 500 $xg$ and filter using a 100 μm cell strainer.

Whole blood was retrieved from the femoral or carotid arteries. At least 100μl of sample was collected in 1ml of PBS + 5 mM EDTA to prevent clotting. Red blood cells were lysed with 5 min incubation in ACK buffer and then 5ml of cDMEM was added to the cells. Media was removed by centrifugation at 500 $xg$ for 7 min at 4°C and then resuspended in FACS buffer for antibody staining for flow cytometry.

Isolation of cells from mediastinal lymph nodes (lung draining lymph nodes) was performed by mashing the organs in 100 μm cell strainers. Homogenates were then treated with ACK buffer for 3 min and washed once with cDMEM. Cells were resuspended in FACS buffer for antibody staining as described above.

### Flow cytometry

$2.5 \times 10^6$ cells isolated from the lung were treated with purified rat IgG2b anti-mouse CD16/CD32 receptor antibody (clone 93) for 20 min at 4°C (Biolegend, Cambridge, UK). RSV-specific CD8+ T cells were stained for 30 min at room temperatures using Alexa Fluor 647-conjugated $M_{187-195}$ tetramers (H-2D$^b$/ NAITNAKII) obtained from the NIH Tetramer Core Facility (Emory University Atlanta, GA, USA). Cells were stained with fluorochrome-conjugated antibodies against CD3 (17A2, AF700), CD4 (GK1.5, PE or BV786), CD8 (53–6.7, eFluor780), CD19 (6D5, FITC or PE-CF594), CD44 (IM7, PE-Cy7), CD45 (30-F11, BV605), CD62L (MEL-14, BV421), Ly6G (1A8, BV570), Ly6G (1A8, FITC), CD69 (H1.2F3, BUV737), CD103 (2E7, PerCP-Cy5.5), CD49a (Ha31/8, BUV395 or PE), CXCR6 (SA051D1, BV711 or APC), Siglec F (E50-2440, BV786), CD11c (HL3, V450), CD11b (M1/70, AF700) and CD64

(X54-5/7.1, APC) in PBS with fixable live-dead Aqua dye (Invitrogen, Paisley, UK) for 30 min at 4˚C before fixing the cells with fixation buffer (Biolegend, Cambridge, UK). All antibodies were purchased from BD, eBioscience or Biolegend.

To detect intracellular cytokines, lung and BAL isolated cells were stimulated for 4h with 5 μg/ml RSV $M_{187-195}$-peptide at 37˚C. After 1h of incubation Golgi Plug (BD Biosciences) was added 1 μl per $2.5 \times 10^6$ cells according to manufacturer's instructions and incubated for 3h. Cells were stained for surface markers as described above and then fixed. Then cells were stained with fluorochrome-conjugated antibodies against granzyme B (GB11, PE-CF594) and IFN-γ (XMG1.2, BV711) in the presence of purified rat IgG2b anti-mouse CD16/CD32 receptor antibody in permeabilization buffer (BioLegend) for 1h. For Ki67 intranuclear staining, previously surface stained cells were treated with BD FoxP3/transcription factor working solution for an hour and then stained with APC-conjugated anti-Ki67 antibody (16A8) in the presence of purified rat IgG2b anti-mouse CD16/CD32 receptor antibody in permeabilization buffer for 1h. Samples were measured on a Becton Dickinson Fortessa LSR-SORP equipped with 20mW 355nm, 50mW 405nm, 50mW 488nm, 50mW 561nm, 20mW 633nm lasers and a ND1.0 filter in front of the FSC photodiode. For acquisition, PMT voltages where set after CST standardized checks to maximize data precision and 250,000 single live CD45+ events were recorded. Data were analyzed using FlowJo software (Treestar, Ashland, OR, USA). The automated analysis in Fig 4 was performed in a concatenated file generated with 105.000 CD8+ $M_{187-195}$ tetramer+ events downsampled from 4 wt, 5 *Mavs*<sup>-/-</sup> and 5 *Myd88/Trif*<sup>-/-</sup> mice (7.500 events per animal). Automated analysis in Fig 5 was performed in a concatenated file generated with 350.000 CD8+ events downsampled from 4 wt, 5 *Mavs*<sup>-/-</sup> and 5 *Myd88/Trif*<sup>-/-</sup> mice (25.000 events per animal). Uniform Manifold Approximation and Projection (UMAP v3.1) used for dimensionality reduction and the FlowSOM (v2.6) algorithm [70] for automatic cluster identification.

## Cytokine detection

BAL supernatants were assessed for IFN-γ and granzyme B using ELISA kit (R&D Systems, Minneapolis, MN, USA). IFN-α was detected by ELISA [32]. Detection limits were 31 pg/ml for IFN- γ, 16 pg/ml for GzmB and 150 IU/ml for IFN-α.

## RNA isolation and quantitative RT-PCR

Lung lobes were homogenized using a TissueLyser LT (Qiagen). Total RNA was extracted from homogenized lung tissue using RNeasy Mini kit including DNA digestion as described by the manufacturer (Qiagen). One μg of purified RNA was transformed to cDNA using High-Capacity RNA-to-cDNA kit (Applied Biosystems, Paisley, UK) according to manufacturer's instructions. Quantitative RT-PCR reaction for *Ifng* and RSV L gene was performed using primers and probes as previously described [43] in QuantiTect Probe PCR Master Mix (Qiagen). Exact copy number was obtained using a plasmid standards and results were normalized to *Gapdh* expression (Applied Biosystems). The relative expression of *Ifnb*, *Ccl2*, *Cxcl1*, *Cxcl9* and *Cxcl10* (all from Applied Biosystems) to the housekeeping gene *Gapdh* was determined. Difference of cycle threshold (ΔCt) to *Gapdh* was calculated and results are reported as $2^{-\Delta Ct}$. Analysis was performed using 7500 Fast System SDS Software (Applied Biosystems).

## Statistical analysis

For simple two-group comparison, unpaired two-tailed Student's t test. For multiple comparisons One or two-way ANOVA was used following Tukey's post hoc test. For all tests, a value

of P<0.05 was considered as significant. *p<0.05; **p<0.01; ***p<0.001. Statistical analysis of data was performed using GraphPad Prism 7 (GraphPad Software Inc., La Jolla, CA, USA).

## Supporting information

**S1 Fig. Gating strategy to identify cell populations in lungs and airway during RSV re-infection.** Lung cells were obtained after collagenase digestion and BAL cells obtained and stained for different surface and intracellular markers. Representative plots from lung cells are shown. (**a**) Flow cytometry analysis was performed on 250,000 CD45+ events after excluding debris, doublets and dead cells. (**b**) Gating strategy used to identify neutrophils, alveolar macrophages (AMs), naïve CD8+ and CD4+ T cells, CD8+ and CD4+ $T_{EF}$ cells, CD8+ $T_{RM}$ cells and IFN-γ and GzmB producing CD8+ $T_{RM}$ cells. (**c**) Strategy used for identification of vasculature (i.v.+) or lung resident (i.v.-) leukocytes using i.v. *in vivo* staining with anti-CD45 BUV394. (**d**) Gating strategy used to identify $M_{187-195}$ specific CD8+ T cells. All gates were defined using fluorescence minus one (FMO) controls for each antibody used. The same gating strategy was used in airway cells purified from bronchoalveolar washes.
(TIF)

**S2 Fig. Immune cell populations in the lung and airways following RSV secondary challenge.** Mice were RSV infected i.n.. At 3 weeks p.i. mice were mock or re-challenged with RSV (day 0 = mock re-infection) and lung and airway cells were analyzed at the indicated time-points by flow cytometry. Total lung and airway (**a**) CD45+ cells, (**b**) neutrophils, (**c**) alveolar macrophages (AMs), (**d**) total CD8+ T cells, (**e**) total CD4+ T cells, (**f**) effector (CD62L- CD44+) CD4+ T cells, (**g**) lung CD8+ and CD4+ naïve T cells, and (**h**) lung and airway CD69+ CD4+ effector (CD62L- CD44+) T cells were quantified by flow cytometry. Data are presented as the mean ±SEM of 9–11 individual mice per time point, pooled from two independent experiments. Statistical significance of differences between day 0 (mock re-infected) and other time points was determined by one-way ANOVA with Tukey's post hoc test. * indicates differences between day 0 and days 1–4. * P ≤ 0.05, ** P ≤ 0.01, *** P ≤ 0.001.
(TIF)

**S3 Fig. CD8+ tissue resident memory T cells expansion is antigen specific.** RSV infected mice were intranasally re-infected with RSV or infected with influenza A virus (X31 strain; 250 PFU per mice). Four days later mice were euthanized and (**a**) $T_{RM}$ cells and (**b**) $M_{187-165}$ specific $T_{RM}$ cells were quantified in lungs and BAL. (**c**) Representative histograms showing Ki67 expression on CD8+ T cells in lung at different days post RSV re-infection. Vasculature CD45 cells were labeled *in vivo* by intravenous injection of anti-CD45 antibody and i.v. CD45- (**d**) CD49a+ and (**e**) CXCR6+ $M_{187-195}$+ cells were quantified in lungs at different days post re-infection. (**f**) i.v. CD45- $M_{187-195}$ specific CD8+ T cells and $T_{RM}$ cells were quantified in lung draining lymph nodes (LN) after RSV re-infection. Panels a, b, c show data of one representative experiment. In panels d, e and f data are presented as the mean ±SEM of 9–11 individual mice per time point, pooled from two independent experiments. Statistical significance was determined by one-way ANOVA with Tukey's post hoc test. In panels d, e, f * indicates differences between day 0 and days 1–4. * P ≤ 0.05, ** P ≤ 0.01, *** P ≤ 0.001.
(TIF)

**S4 Fig. Vascular and resident lung cells after FTY720 treatment during RSV secondary challenge.** Mice re-infected with RSV were treated with 25 µg FTY720 i.p. administered daily from day -2 prior to re-challenge until day 3 post RSV re-challenge. Mice were given 2µg of CD45-BUV395 i.v. 10 min prior to euthanasia to distinguish cells in the vasculature of the lung from resident lung cells. (**a**) Alveolar macrophages (AMs), (**b**) neutrophils, (**c**) total

CD4+, (**d**) naïve CD4+ and (**e**) effector (CD62L- CD44+) CD4+ T cell, (**f**) total CD8+, (**g**) naïve CD8+, and (**h**) effector (CD62L- CD44+) CD8+ T cell were assessed by flow cytometry discriminating between vascular (i.v. CD45+) and resident (i.v. CD45-) populations. Blood samples were obtained immediately post-mortem from the femoral vein and (**i**) neutrophil, (**j**) CD8+ and (**k**) CD4+ T cells were quantified by flow cytometry. Data are presented as the mean±SEM of 7 PBS re-infected (RSV/PBS), 12 vehicle-treated and 9 FTY720-treated re-infected individual mice pooled from two independent experiment. Statistical significance of differences between groups was determined by one-way ANOVA with Tukey's post hoc test. $^*$ P $\leq$ 0.05, $^{**}$ P $\leq$ 0.01, $^{***}$ P $\leq$ 0.001.
(TIF)

**S5 Fig. Effect of MAVS and MyD88/TRIF deficiency in RSV associated weight loss and CD8+ T cell memory response during RSV re-challenge.** (**a**) wt, *Mavs*$^{-/-}$ and *Myd88/Trif*$^{-/-}$ mice were infected i.n. with RSV. Body weight was monitored throughout primary infection and percentage of original weight was quantified. (**b**) Lung cells were recovered stained with tetramers loaded with M$_{187-195}$ RSV peptide to quantify the percentage of RSV-specific cells within the CD8+ T$_{RM}$ cells after primary infection. Mice were re-infected with RSV and 4 days later(**c**) total and (**d**) RSV-specific CD8+ T cells were quantified. Lung and airway cells were stimulated with RSV M$_{187-195}$ peptide and IFN-γ and GzmB production was detected by intracellular staining and quantified in CD8+ T$_{EM}$ cells (CD62- CD44+) cells using flow cytometry. Total number of IFN-γ and GzmB positive CD8+ CD69+ CD103+ T$_{RM}$ cells in (**e**) lung tissue and (**f**) airways. (**g**) Number of T$_{RM}$ cells was quantified in the absence and presence of M$_{187-195}$ peptide stimulation (**h**) RSV M$_{187-195}$ peptide stimulated CD8+ T cells were analyzed using dimensionality reduction software (UMAP) and automatic cluster identification (FlowSOM). FlowSOM tree-plots for each genotype are presented, showing pie charts with marker expression for each node and color identification of each of the 8 identified populations (corresponding to populations shown in Fig 5G). Pie chart diameter represents the proportion of each node in the data set. Red arrow indicates major change between wt, *Mavs*$^{-/-}$ and *Myd88/Trif*$^{-/-}$ mice. *Mavs*$^{-/-}$ mice were treated i.n. at 6h and 18h during primary RSV infection with 500ng of recombinant IFN-α. Four days after re-infection mice were euthanized and (**i**) M$_{187-195}$ specific T$_{RM}$ cells were quantified in the lungs, and GzmB+ and IFN-γ+ T$_{RM}$ cells were quantified in *ex vivo* M$_{187-195}$ peptide stimulated (**j**) lung and (**k**) BAL cells. In a, data are presented as the mean±SEM of 19 wt, 19 *Mavs*$^{-/-}$ and 19 *Myd88/Trif*$^{-/-}$ mice pooled from three independent experiment. Statistical significance of differences between groups was determined by two-way ANOVA with Tukey's post hoc test. In b-d, data represents the mean±SEM of 9 wt, 11 *Mavs*$^{-/-}$ and 10 *Myd88/Trif*$^{-/-}$ mice pooled from two independent experiment. Panel i show data of one experiment with 4–5 mice per group. $^*$ indicates differences between wt and *Mavs*$^{-/-}$ and # differences between *Mavs*$^{-/-}$ and *Myd88/Trif*$^{-/-}$(in panel a). $^*$ P $\leq$ 0.05, $^{**}$ P $\leq$ 0.01, $^{***}$ P $\leq$ 0.001.
(TIF)

**S6 Fig. Immune cell infiltration and viral control during RSV re-infection of wt, *Mavs*$^{-/-}$, IFN-α treated *Mavs*$^{-/-}$ and *Myd88/Trif*$^{-/-}$ mice.** RSV infected wt, *Mavs*$^{-/-}$, *Myd88/Trif*$^{-/-}$ mice and IFN-α treated *Mavs*$^{-/-}$ mice were re-infected with RSV and euthanized at day 2 and 4 post re-infection. (**a**) Viral load was determined in lung tissue by L-gene detection by qPCR. Lung cells were recovered and (**b**) total cell count, (**c**) neutrophils and (**d**) CD64+ inflammatory monocytes were quantified in the lung by flow cytometry. BAL cells were recovered and (**e**) total cell count, (**f**) neutrophils and (**g**) CD64+ inflammatory monocytes were quantified in the lung by flow cytometry. *Cxcl10*, *Ccl2* and *Cxcl1* expression levels in lung tissue were determined by qPCR. Data are presented as the mean±SEM of 9–11 mice per group pooled from two independent experiment. Statistical significance of differences between groups was

determined by one-way ANOVA with Tukey's post hoc test. * indicates differences between groups. * P ≤ 0.05, ** P ≤ 0.01, *** P ≤ 0.001.
(TIF)

**S1 Table. Antibodies and other reagents used for flow cytometry for the murine studies.**
(XLSX)

## Acknowledgments

We thank S. Akira (World Premier International Immunology Frontier Research Center, Osaka University, Osaka, Japan) for providing *Mavs*$^{-/-}$ and *Myd88/Trif*$^{-/-}$ mice. We thank Christopher Chiu and Caetano Reis e Sousa for critically reading the manuscript. We thank the NIH Tetramer Core Facility for RSV $M_{187-195}$ tetramer (H-2D$^b$/NAITNAKII). We also thank James Harker for the protocol for FTY720 treatment, Yanping Guo and Rhadika Patel of the St Mary's flow cytometry facility and the staff of the St Mary's animal facility for their assistance.

## Author Contributions

**Conceptualization:** Augusto Varese, Cecilia Johansson.

**Data curation:** Cecilia Johansson.

**Formal analysis:** Augusto Varese, Joy Nakawesi, Cecilia Johansson.

**Funding acquisition:** Augusto Varese, Cecilia Johansson.

**Investigation:** Augusto Varese, Joy Nakawesi, Ana Farias, Freja C. M. Kirsebom, Michelle Paulsen, Rinat Nuriev, Cecilia Johansson.

**Project administration:** Cecilia Johansson.

**Writing – original draft:** Augusto Varese, Cecilia Johansson.

**Writing – review & editing:** Augusto Varese, Joy Nakawesi, Ana Farias, Cecilia Johansson.

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
