## [Decision Letter · Decision Letter 0]

9 Aug 2021

Dear Dr. Johansson,

Thank you very much for submitting your manuscript "Type I interferons and MAVS signaling are necessary for tissue resident memory CD8+ T cell responses to RSV infection" for consideration at PLOS Pathogens. As with all papers reviewed by the journal, your manuscript was reviewed by members of the editorial board and by several independent reviewers. In light of the reviews (below this email), we would like to invite the resubmission of a significantly-revised version that takes into account the reviewers' comments.

We cannot make any decision about publication until we have seen the revised manuscript and your response to the reviewers' comments. Your revised manuscript is also likely to be sent to reviewers for further evaluation.

Sincerely,

Kanta Subbarao

Section Editor

PLOS Pathogens

Kanta Subbarao

Section Editor

PLOS Pathogens

Kasturi Haldar

Editor-in-Chief

PLOS Pathogens

orcid.org/0000-0001-5065-158X

Michael Malim

Editor-in-Chief

PLOS Pathogens

orcid.org/0000-0002-7699-2064

Reviewer's Responses to Questions

**Part I - Summary**

Reviewer #1: The manuscript by Varese et al., explores the role of innate immune components in the generation and re-call of lung Trm following RSV infection. They show that mice lacking key components of innate signalling pathways (MAVS) display an impaired local expansion and activation of lung Trm following RSV challenge. Interestingly, the introduction of type I interferon in MAVS mice during a primary RSV infection resulted in the development of lung Trm with improved capacity to undergo secondary recall expansion although these cells still retained defects in recall effector function. This is an interesting and well written study supported by robust data. However, one major caveat present throughout the paper is that in most experiment profiled bulk CD8+ memory T cells instead of RSV specific cells. It does appear the group has access to RSV tetramers and therefore I strongly recommend that they go back and repeat some of the key experiments tracking the virus specific cells which would provide more informative information than profiling the bulk memory T cell pool.

Reviewer #2: This manuscript examines the role of type I IFNs and MAVS signaling in the generation and effector functions of tissue resident memory CD8 T cells specific to respiratory syncytial virus (RSV) in mice. The authors perform initial kinetics of the Trm response after RSV re-infection including an examination of granzyme B expression and IFN-gamma production capacity by the cells following peptide restimulation. Additional experiments examining the response in MAVS or Myd88/Trif deficient mice suggest that MAVS plays an important role in the expansion and effector functions of the Trm following RSV reinfection. Additional experiments indicate tat type IFNs during the primary RSV infection may be critical to the initial generation of the Trm cells in the lung. The manuscript is well written and the experiments are generally well designed. However, a major limitation of the manuscript is the failure to use the widely utilized in vivo labeling approach to distinguish cells within the lung tissue from those in the vascular space which is likely to impact the interpretation of a number of the critical experiments performed at day 4 after virus rechallenge when there will be many cells trafficking towards the lung in the blood that may not be in the lung tissue at that timepoint.

Reviewer #3: In this manuscript the studies have addressed the presence of Trm cells with clearance of RSV upon reinfection. These data are important since many studies have identified Trm cells as critical for the long term protection of re-infection. Trm cells are most often examined within the tissue and are present upon the reinfection to quickly respond to the pathogen locally. These studies have further suggested that MAV mediated activation is important for the generation of Trm cells upon reinfection. The studies also suggested that the presence of type I IFN during the primary infection was critical for reconstituting the "protective" reinfection response. These conclusions are logical and follow previous data from this and other groups.

**Part II – Major Issues: Key Experiments Required for Acceptance**

Reviewer #1: 1. Fig 1a-f –The analysis of Trm numbers following re-infection should be restricted to antigen specific tetramer+ cells.

2. Fig 1d – It would be informative if the authors could compare the numbers of effector T cells seen following reinfection to that seen during a primary infection

3. Fig 1e - It has been previously documented by others that Trm undergo local recall expansion in tissues following reinfection. Can the authors show whether there is also evidence of expansion in the lung draining LN. It would be interesting to see whether there is a difference in kinetics of recall at local (lung) and distal (LN) sites.

4. Fig 1a-d - Is the expansion of CD8 T cells in the lung following secondary infection with RVS antigen specific – would infection with a different respiratory pathogen (ie influenza) also see rapid increase in Trm numbers?

5. Supplementary Fig 2 –It would be informative to compare the kinetics of immune cell infiltration into the lung following secondary infection to that observed during primary infection

6. Fig 2a - Is the rapid clearance of RSV following secondary exposure mediated by the local T cells. Would CD8 or CD4 T cell depletion prior to secondary infection impact viral control? Also, it would be informative to compare these titres to those seen after a primary infection

7. Fig 2e-g – The authors are using CD69+CD103 expression to identify Trm in these ex vivo peptide stimulation assays. Does peptide stimulation ex vivo impact the expression of these markers (and therefore affect the interpretation of this assay). Can the authors show the frequency of CD69+CD103+ T cells in these assays with and without peptide stimulation – if peptide stimulation is not altering expression, these should match.

8. Fig 3 – It would be more informative if tetramer+ cells were tracked in this experiment

9. Fig 4c – Can the authors show fold change in Trm numbers post re-infection in WT, MAVS-/- and Myd88/Trif KO. It is hard to compare defects in expansion when there are differences in starting numbers of Trm. Once again – it would be useful to show tetramer+ cells rather than bulk CD8 T cells in this experiment.

10. Fig 4i – was UMAP analysis performed on tetramer+ cells? If not, it would interesting to compare the profile presented to that of the tetramer+ cells.

11. To show that the impaired Trm pool generated following primary infection of MAVS-/- mice has a biological impact on the capacity to control a secondary infection viral loads in the lung of reinfected wt and MAVS-/- mice should be compared. Moreover, it would be interesting to see whether IFNa treatment of MAVS mice during the primary infection impacts the capacity of the animals to control a secondary infection. Do viral loads in the lung following re-infection of these mice differ from MAVS-/- not subjected to IFNa treatment?

Reviewer #2: 1. A major limitation of the study is that the authors have not used in vivo antibody labeling to distinguish cells in the lung tissue from the peripheral blood. This is a major issue when performing these types of studies as many memory cells with similar phenotypes can be found in the blood and not present in the lung tissue. This is a major concern particularly when examining the memory response in RSV immune mice re-challenged with RSV and looking at day 4 as at this time point many cells will be trafficking towards the lung and it is important to distinguish the cells in the lung tissue from those in the peripheral blood to properly interpret the data and fully support the conclusions the authors attempt to make from their results. Perfusion of the lung is not adequate to remove the cells that will be in the small capillary beds of the lung which can result in an artificially high estimate of the cells in the lung tissue.

2. Total numbers of cells should be reported in Figure 5b-c in addition to the frequencies as in the other figures. This is a critical omission necessary to interpret the data presented and support the conclusions of this key point in the paper.

3. Similar to point 2 above for Figure 6h-i.

Reviewer #3: 1. The concept of Trm cells is that they are resident in the tissue prior to a re-infection challenge and protect the host from the response. While the cell surface phenotype of the Trm cells were examined, they were primarily detected after 4 days of re-infection. Can these truly be considered Trm cells? How long after 4 days do these cells persist in the lung? By definition these cells exist in tissue and do not recirculate.

2. The primary endpoint that is examined, especially in the Mavs deficient animal studies is weight loss. The studies should examine additional endpoints, including viral clearance and inflammation/histopathology. The weight loss measurement has been primarily shown to be accompanied by TNF and other innate cytokines. What is the cellular infiltration in the lungs at 8 days post-infection when the peak pathology (i.e. weight loss) occurs? Are there increased/decreased numbers of granulocytes, eosinophils and neutrophils?

3. An additional parameter that might be helpful in these studies would be lung physiology. Are there functional consequences to the differences in responses within the different groups at anytime during the re-infection?

4. While the data clearly demonstrates that there are secondary infection consequences that are occurring in the Mavs deficient mice, the characterization of what are the responses in the primary responses with or without type I IFN would be important. What are the changes in the primary immune response when Mavs is absent and when type I IFN is reconstituted?

**Part III – Minor Issues: Editorial and Data Presentation Modifications**

Reviewer #1: (No Response)

Reviewer #2: 1. It is unclear why there is twice as many IFN-g producing CD8 Trm cells between Fig 2 E at day 4 and Fig 5a at day 4 which should essentially be the identical conditions.

2. Examination of total GzmB protein levels and Ifng RNA levels are not very informative since it is much more important to know the changes in both the frequency and total number of cells capable of making these molecules.

Reviewer #3: (No Response)

PLOS authors have the option to publish the peer review history of their article (what does this mean?). If published, this will include your full peer review and any attached files.

Reviewer #1: No

Reviewer #2: No

Reviewer #3: No
---

## [Decision Letter · Decision Letter 1]

12 Jan 2022

Dear Dr. Johansson,

We are pleased to inform you that your manuscript 'Type I interferons and MAVS signaling are necessary for tissue resident memory CD8+ T cell responses to RSV infection' has been provisionally accepted for publication in PLOS Pathogens.

Best regards,

Kanta Subbarao

Section Editor

PLOS Pathogens

Kanta Subbarao

Section Editor

PLOS Pathogens

Kasturi Haldar

Editor-in-Chief

PLOS Pathogens

orcid.org/0000-0001-5065-158X

Michael Malim

Editor-in-Chief

PLOS Pathogens

orcid.org/0000-0002-7699-2064

The reviewers appreciated the additional data in the revision.

Reviewer Comments (if any, and for reference):

Reviewer's Responses to Questions

**Part I - Summary**

Reviewer #1: The authors have performed a very comprehensive and thorough revision providing additional data as supplementary figures which have been appropriately highlighted in the text of the manuscript. The additional experiments have certainly strengthened the main findings in the manuscript.

Reviewer #3: (No Response)

**Part II – Major Issues: Key Experiments Required for Acceptance**

Reviewer #1: (No Response)

Reviewer #3: (No Response)

**Part III – Minor Issues: Editorial and Data Presentation Modifications**

Reviewer #1: (No Response)

Reviewer #3: (No Response)

PLOS authors have the option to publish the peer review history of their article (what does this mean?). If published, this will include your full peer review and any attached files.

Reviewer #1: No

Reviewer #3: No

---

## [Editor Report · Acceptance letter]

28 Jan 2022

Dear Dr. Johansson,

We are delighted to inform you that your manuscript, "Type I interferons and MAVS signaling are necessary for tissue resident memory CD8+ T cell responses to RSV infection," has been formally accepted for publication in PLOS Pathogens.

Best regards,

Kasturi Haldar

Editor-in-Chief

PLOS Pathogens

orcid.org/0000-0001-5065-158X

Michael Malim

Editor-in-Chief

PLOS Pathogens

orcid.org/0000-0002-7699-2064